# Reduced Toxicity of *Centruroides vittatus* (Say, 1821) May Result from Lowered Sodium β Toxin Gene Expression and Toxin Protein Production

**DOI:** 10.3390/toxins13110828

**Published:** 2021-11-22

**Authors:** Aimee Bowman, Chloe Fitzgerald, Jeff F. Pummill, Douglas D. Rhoads, Tsunemi Yamashita

**Affiliations:** 1Department of Biological Sciences, Arkansas Tech University, 1701 N Boulder Ave., Russellville, AR 72801, USA; aimeebowman13@gmail.com (A.B.); chughes11@atu.edu (C.F.); 2High Performance Computing Center, University of Arkansas-Fayetteville, 850 W Dickson St., Fayetteville, AR 7270, USA; jpummil@uark.edu; 3Department of Biological Sciences, University of Arkansas-Fayetteville, 850 W Dickson St., Fayetteville, AR 7270, USA; drhoads@uark.edu

**Keywords:** *Centruroides*, sodium β toxin, RT-qPCR, transcriptome, proteome

## Abstract

Body tissue and venom glands from an eastern population of the scorpion *Centruroides vittatus* (Say, 1821) were homogenized and molecular constituents removed to characterize putative sodium β toxin gene diversity, RT-qPCR, transcriptomic, and proteomic variation. We cloned sodium β toxins from genomic DNA, conducted RT-qPCR experiments with seven sodium β toxin variants, performed venom gland tissue RNA-seq, and isolated venom proteins for mass spectrophotometry. We identified >70 putative novel sodium β toxin genes, 111 toxin gene transcripts, 24 different toxin proteins, and quantified sodium β toxin gene expression variation among individuals and between sexes. Our analyses contribute to the growing evidence that venom toxicity among scorpion taxa and their populations may be associated with toxin gene diversity, specific toxin transcripts variation, and subsequent protein production. Here, slight transcript variation among toxin gene variants may contribute to the major toxin protein variation in individual scorpion venom composition.

## 1. Introduction

Scorpions of the genus *Centruroides* constitute the most medically important and one of the most diverse and wide-ranging scorpion taxa in North America [1,2]. The complex evolutionary history and geographic variability of this genus has generated controversy and taxonomic confusion [2,3,4,5]. This complex evolutionary history also appears in the diversity of the venom components. Recent studies show intraspecific complexity and diversity in the protein and peptide constitution of the venom components [6,7]. In addition, age-specific and sex-specific variation has been reported in *Centruroides vittatus* and *C. hentzi* [8,9,10]. These studies suggest that a rich evolution of venom components as *Centruroides* species and their populations expanded and diversified into their present geographic range.

Scorpion venom toxicity results from a wide range of venom components (peptides, lipids, nucleotides, and other compounds) [11]. Most research has focused upon characterizing proteins and gene activity via proteomic and transcriptomic studies [10,12,13]. These proteins have been shown to affect sodium, potassium, chloride, and calcium ion channels primarily in muscle or nervous tissue [11,12]. Voltage-gated sodium channels (VGSCs) are critical to initiate action potentials in excitable cells [14,15]. Scorpion toxin proteins are especially interesting as they have the ability to discriminate among subtypes of the phylogenetically conserved sodium channels, which are difficult to discriminate with most drugs [15,16]. Two classes exist with scorpion sodium toxins, an α toxin class and a β toxin class [11,12]. Although these toxins affect receptor sites on the sodium channel external pore surface (gating module) rather than the pore, both target distinct receptor sites [11,12]. The α class toxins are those that inhibit sodium channel inactivation through binding at receptor site 3. These α toxins bind sodium channels open, which allows sodium influx to compromise cell activity. The β toxins act as sodium channel modifiers, through shifting the voltage dependence of activation in a negative direction promoting spontaneous and repetitive sodium channel opening after binding to receptor site 4 allowing sodium influx [14,17]. Studies have shown that scorpion sodium β toxins affect sodium channel voltage sensors that respond to membrane depolarization [18].

Research has linked amino acid residue conservation at specific sites with toxin activity [11,12]. Other studies have conducted mutagenesis at specific sites to create toxins with reduced activity at sodium channels [11,19]. Minute changes in toxin proteins such as amino acid substitutions, deletions and modifications at short protein sequences have resulted in higher toxicity to mice [11,20]. Other recent studies have illustrated similar functional surface anatomy of β toxins in their alpha helix and at the C-terminus that confer interactions with sodium channel sites [18]. In addition, removal of residues at the N and C terminus in β toxins resulted in a non-toxic, truncated 46AA protein, unable to bind to its original sodium channel receptor but yet showed activity with the sodium channel [19]. Furthermore, more recent data have shown a multitude of variation between α and β forms and suggest that undescribed unique structural and functional relationships exist [11,12,14].

The scorpion *Centruroides vittatus* encompasses a large geographic range across the western USA and northern United Mexican States (Figure 1). Although a member of the toxic *Centruroides* genus, this species is not known as medically important [21]. However, due to coevolution with mammalian predators, evidence suggests that western *C. vittatus* populations may possess a more medically significant venom than eastern populations [22]. Throughout its geographic range, *C. vittatus* is commonly found in diverse ecological habitats, but in populations across the northern and eastern geographic distributions it appears to prefer dry, rocky south facing slopes or glade areas. Human introduction of this scorpion appears to also have created additional populations outside its known geographic range [23].

We hypothesize that potentially limited sodium β toxin population gene diversity and gene transcript variation along with a decreased proteome diversity and abundance may contribute to a reduced medically significant venom in this scorpion. Our findings of putative sodium β toxins and similar peptides suggest that genomic, transcriptomic, and proteomic components contribute to venom variation among populations potentially resulting in a reduced medically significant venom when compared to other medically important *Centruroides* species.

## 2. Results

### 2.1. Sodium β Toxin Variability Analysis

Through the sodium β toxin cloning experiment, we identified a total of 79 putative *C. vittatus* sodium β toxins with 72 putative new sodium β toxins and select toxins presented in Figure 2. All toxin variants are shown in Appendix A. Sixteen identified sequence variants may represent pseudogenes as they housed internal stop codons or stop codons that created truncated proteins. Two toxins were commonly identified across eight and six populations (Table 1). In general, we show that western and southern populations (e.g., AgSp and Kisa) populations show greater sodium toxin gene diversity than northern and eastern populations (e.g., LBR and KS) (Appendix A).

### 2.2. RT-qPCR Experiments

The data from the RT-qPCR experiments suggest that no significant expression difference occurs among males; however, there was high variation among samples (Figure 3, Table 2, Appendix A). Females exhibited few significant differences, yet sodium toxin Ha1210, unique to eastern populations, showed a higher expression value than all other sampled sodium toxin genes, with significant differences to Chin654, AgSp668, and CvIV4. These data suggest that the pain-inducing toxin, CvIV4, is expressed in lower amounts and may contribute to the lower pain associated with eastern scorpions.

### 2.3. Transcriptome

The basic transcriptome contig statistics are shown in Table 3. The BLASTn query identified 1491 unique hits (Blastgrabber) with 111 total toxin gene ids and 26 sodium toxin genes (Table 4). The blast query of the transcriptome with the NCBI 2133 toxin gene sequences produced 1142 significant alignments with the following toxin categories (Appendix A): NaScTx (19), Metalloproteases (10), KScTx (6), Phospholipases (2), IGFBP (2), Host Defense Peptides (2), and Serpins (1). We also discovered eleven additional sodium β toxin transcripts in our transcriptome of an eastern population when compared to the five revealed in the cloning experiment for toxin variability from the same regional population.

The summary RPKM values for the two adult males and one female support the RT-qPCR experiments with higher expression values for sodium β toxins vs. the pain inducing Cv Alpha IV4 toxin (Table 4). In addition, the reference genes chosen for the RT-qPCR experiments show little deviation between body tissue and venom glands (i.e., EF1, EF2, RPL19).

### 2.4. Toxin Proteomics

The proteomic analysis from the telsons of three males and four females resulted in 273 proteins identified in the total spectra with 24 different toxin proteins and nine different sodium β toxins (Table 5 and Appendix A). Among the sodium β toxins, CviNaTBet_Ha1210 exhibits the highest total abundance with the two other sodium β toxins targeted in the RT-qPCR experiments exhibiting much lower values. The sodium β toxins identified from other species show lower NSAF values than putative sodium β toxins in these individuals.

## 3. Discussion

### 3.1. Sodium β Toxin Gene Diversity

Our sodium β toxin diversity analysis shows marked variability among populations in toxin gene diversity. Western populations exhibit >10 putative variants, whereas eastern populations (e.g., KS) exhibit five putative toxin variants. Western and southern populations may exhibit higher toxin gene diversity, presumably due to longer population isolation and separation. A phylogeographic analysis of *C. vittatus* populations showed western and southern populations experienced longer divergence times than those populations in the eastern and northern portions of their geographic range [24]. In addition, western scorpion populations overlap with grasshopper mice populations, a key predator of scorpions. These scorpion populations appear to have coevolved a more painful mammalian toxin to deter mice predation with subsequent increased pain tolerance from grasshopper mice [22].

Interestingly, our analysis of toxin gene variants suggests that pseudo genes may exist in the genome with increased numbers identified from samples in western populations (Appendix A). Many of the sequence variants coded for polypeptides with the number of AA residues similar to that seen in known variants; however, nine putative toxin variants show increased length to ~80 AAs, with six showing 10 cysteine residues instead of the 8 typically noted for β toxins and may represent another class of venom toxins. The additional cysteine residues are unique to this study but these variants appear to show AA homology with shorter known toxin variants. As this dataset did not separate mammalian and insect toxins, nor demonstrate physiologically alpha versus beta toxins, the existence of these putative toxin proteins will need to be confirmed through further proteomic and transcriptomic analyses of other *Centruroides* and buthid species. Lastly, mild toxicity in other buthid scorpions appears to result from a substitution of the lysine from the first amino acid in the polypeptide [25]. *C. vittatus* populations do not appear to harbor this substitution; therefore, we speculate that differential gene expression explains the lowered toxicity in these populations.

### 3.2. Sodium β Toxin Gene Expression Variation

Our gene expression analysis via RT-qPCR shows more variation in venom gene activity in males than females as indicated by the normalized relative expression values (Figure 3). This result is echoed in the results from a proteomic analysis of venom components in populations of *C. hentzi* [7]. Male *C. vittatus* scorpions move more extensively than females and venom diversity may be a byproduct of male movement as males encounter more microclimates and potential selection pressures for venom variation [7,8,26]. The 13 females show the highest normalized relative expression values for the putative toxin CviNaTBet_Ha1210, (Figure 3) and this result is also seen in the RPKM values in the transcriptome contig assembly and the NSAF percentages in the protein abundance (Table 4 and Table 5). The pain-inducing toxin, Cv Alpha Tox IV4, shows a lower normalized relative expression value along with four other toxins, and also exhibits low values in the transcriptome contig assembly and the NSAF percentages in the protein abundance. These results may explain how a less painful toxin occurs in this species.

### 3.3. Sodium β Toxin Transcriptome and Proteome

Our transcriptomic and proteomic data align with other studies of *Centruroides* venom transcript and proteomic studies showing the majority of toxin transcripts are those of sodium toxins [7,9,10]. If sodium toxin transcripts and protein diversity are compared to each other, we show 19 transcripts but only seven sodium toxin variants in the proteome. This finding is also reflected in *C. hentzi*, where 36 sodium toxin transcripts were revealed but nine sodium toxins were identified in the proteome [10]. Interestingly, a western *C. vittatus* population appears to show similar sodium toxin transcript numbers (13) to our study [9], which suggests that more toxin gene variants may exist in these populations. In addition, our combined datasets indicate preferential expression and production of putative sodium β toxin variants with a reduction in the transcripts and protein abundance of the known pain-inducing toxin, Cv Alpha Tox IV4. However, we note that as entire telsons were assayed for transcriptomic and proteomic analyses, our dataset may reflect lowered toxin transcript and protein abundances. Yet, this trend of greater sodium toxin transcript diversity when compared to sodium toxin protein diversity appears to be reflected in other Buthid species (Table 6).

Our combined dataset indicate that eastern scorpions possess similar sodium toxin gene variants as those in western populations, but may express sodium toxin genes differentially than western populations, especially the known pain-inducing toxin. This conclusion is supported with studies that indicate increased sodium toxin variability may be tied to more severe scorpionism and slight variations in sequence homology can lead to venoms with decreased toxicity [7,32].

## 4. Conclusions

We cloned and identified over 70 putative novel sodium β toxin genes from genomic DNA and showed greater toxin gene diversity may exist in western populations when compared to eastern populations. The RT-qPCR experiments with seven sodium β toxin variants and three reference genes showed sodium β toxin gene expression variation among individuals and indicated males exhibit more variable toxin gene expression than females. The transcriptomic analysis produced 111 toxin gene transcripts with the majority of transcripts identified as sodium β toxin genes. Lastly, the proteomic data suggest a toxin protein reduction from toxin transcript numbers to 24 different toxin proteins. Our analyses correspond to the growing evidence that toxicity in the scorpion genus *Centruroides* may be associated with toxin diversity, specific toxin gene regulation with local variation, and subsequent protein production. In addition, slight variation among toxin variants in *Centruroides* may contribute to the major variation in individual scorpion venom composition.

## 5. Materials and Methods

### 5.1. Population Sodium β Toxin Gene Variability Analysis

We sampled individuals in 11 populations identified from a *C. vittatus* phylogeographic analysis (Figure 1) [24]. Total genomic DNA from up to three scorpion’s pedipalps and a portion of the carapace anterior was extracted with a standard Phenol-Chloroform extraction or with the FastID genomic DNA extraction kit (GeneticIDNA, Inc., Fairfield, IA, USA). After DNA isolation, each sample was further cleaned by Spermine precipitation to optimize subsequent molecular analyses. Extracted genomic DNAs were stored in molecular biology grade water (Sigma Chemical Co., St. Louis, MO, USA) at −20 °C until PCR.

PCR amplification of sodium toxin genes was performed in 25 µL aliquots with the following ingredients: 10 µL total genomic DNA, 2X *Taq* Buffer (150 mM Tris-HCl pH 8.5, 40 mM (NH_4_)_2_SO_4_, 3.0 mM MgCl_2_. 0.2% Tween 20), 1 mM dNTP, 0.5 µM for each primer, 6.25 units *REDTaq* DNA polymerase (Sigma Chemical Co.), 1.6% dimethyl sulfoxide, 0.6% BSA, and 1.6% formamide. The sodium toxin primers were taken from Corona et al. [27]: forward primer 5′-GAGATGAATTCGTTGTTGATGATYA-3′ and reverse primer 5′-GCAATTAAGAAGCGTTACAATA-3′. The cycling conditions consisted of an initial denaturation period of five minutes at 94 °C followed with 30 one-minute cycles of 94 °C, 50 °C annealing, 72 °C extension, and a final seven-minute extension at 72 °C. After PCR products were verified by electrophoresis in a 0.9% agarose gel, the products were GeneCleaned (MP Biomedicals, LLC., Irvine, CA, USA).

Cleaned PCR products were cloned into the pGEM-T vector (Promega, Inc., Madison, WI, USA) with plasmid transformation into *E. coli* JM109 and blue/white selection to identify recombinant clones. From each population, ten random white colonies were selected and archived in glycerol at −80 °C for long term storage. We amplified plasmid inserts from boiled colony samples by PCR with flanking SP6 and T7 Pro primers (Promega, Inc.,). Inserts were identified by agarose gel electrophoresis. Cleaned PCR products were sent to the UAMS DNA Core Sequencing Facility for DNA sequencing with SP6 and T7 Pro primers on an Applied Biosystems 3100 Genetic Analyzer, using the Big Dye Terminator Chemistry, Kit version 1.1 (Foster City, CA, USA). After sequencing, all trace files were reviewed by eye and all ambiguous bases removed from further analysis. Alignment of the sequence data was conducted with Clustal X and Geneious Pro 3.7 [33,34]. After the initial alignment, all toxin DNA sequences were converted into their amino acid sequences to compare to known sodium toxin polypeptides. We named the sodium β toxin sequences with the nomenclature outlines in Romero-Gutierrez et al. [35]. All sequences were deposited in GenBank. Accession numbers are presented in Appendix A and Appendix A.

### 5.2. Sodium β Toxin Gene Expression (RT-qPCR)

We collected male and female scorpions from a population in northwest Arkansas for RNA extraction. Scorpions for all subsequent work were collected from the same population. Scorpions were housed in the laboratory and fed juvenile crickets weekly until harvested for RT-qPCR analysis. Venom was not extracted from scorpions prior to harvesting, thus our samples represent a resting venom gland. Individuals were flash frozen at −80 °C with tissue from the telson (venom gland) and the pedipalp (body tissue) ground for RNA extraction with the Aurum™ Total RNA Mini Kit (BioRad, Inc., Hercules, CA, USA). We determined RNA quality and concentration with a NanoDrop 2000/2000 c (Thermo Fisher Scientific, Inc., Waltham, MA, USA). In addition, RNA integrity was verified via 0.9% agarose gel electrophoresis. We constructed cDNA using an iScript™ cDNA Synthesis Kit (BioRad, Inc.) with incubation parameters of 25 °C for 5 min, 42 °C for 30 min, and 85 °C for 5 min. To verify subsequent RT-qPCR analyses, an initial PCR on the cDNA from both tissues was conducted with RT-qPCR sodium β gene toxin primers and reference gene primers. Cycling parameters were 94 °C for 1 min, 50 °C for 1 min, and then 72 °C for 1 min for 30 cycles, with visualization on a 0.9% agarose gel.

The sodium β toxin gene primers for RT-qPCR were selected from the population sodium β toxin gene survey with six unique to *C. vittatus* (one unique to eastern populations, five in western populations, and *C. vittatus* alpha toxin IV4) and CsEv3b. Sodium β toxin PCR products were verified by sequencing. Reference primers were identified via a blast search of insect reference genes against the *Mesobuthus gibbous* annotated genome. The identified reference genes were then blasted against a *C. vittatus* draft genome assembly (unpublished) to develop reference RT-qPCR primers (Appendix A).

PCR amplification efficiency: A standard curve was created with the following conditions: 5 μL body tissue cDNA diluted to the 5th log in three replicates,10 μL of iTaq™ Universal SYBR^®®^ Green Supermix (BioRad, Inc.), 10 μMol of RPL (Ribosomal protein- L) or EF—1 (Elongation Factor 1) primers, and 4 ul of Sigma-Aldrich molecular biology grade water were added for a total volume of 20 ul per well, and run in a BioRad CFX96 Touch™ Real-Time PCR Detection System with cycling parameters of 95 °C for 30s 95 °C for 5s, 60 °C for 30s with a plate read for 40 cycles.

For Experimental Trials: cDNA from body and telson tissue were analyzed on the nanodrop for concentration and integrity then standardized to 100 ng/ul for a 5 ul of cDNA for all wells except the non-template controls. To all wells, 10 ul of SYBR^®®^ Green was added as well as 1 μL of reference primers (10 μMol) to standard wells and NTCs (Non-Template Controls). One μL (10 μMol) of experimental primer for experimental wells and NTCs, with molecular biology grade water for a final volume of 20 ul to all tubes. Each plate had three replicates of standard and experimental wells and two replicates of NTC wells. The well plate was run in the CFX96 Touch™ Real-Time PCR machine under the same cycling parameters as the standard curve with an added dissociation step for all samples: 95 °C for 10s, 65 °C for 5s with a gradient of 5 to 95 °C. We removed individuals from the experimental dataset if reference and experimental genes showed substantial deviations across individual replicates. The final data included eight males and 13 females.

C_t_ data from triplicate samples were analyzed using the ΔΔCt method [36]. The normalized relative expression values were log2 transformed and post hoc comparisons used one-way ANOVA and Tukey’s pairwise multiple comparison tests.

### 5.3. Transcriptome

Two male and one female scorpion were collected in northwest Arkansas, fed crickets with visual conformation of prey envenomation, then after three days, harvested for telson and carapace transcriptome analysis. The scorpions were flash frozen at −80 °C and total RNA extracted with a Trizol preparation. RNA sample qc was analyzed through electrophoresis with an Aligent TapeStation system. RNA-seq with 50 bp reads was conducted at the University of Delaware on an Illumina genome sequencer (Illumina, Inc., San Diego, CA, USA). The data were viewed for initial quality through FastQC (v0.11.7), trimmed with Trimmomatic (v0.36), and normalization of the data was performed using Trinity (v2.5.1) [37,38,39]. Assembly of the normalized reads was then performed with the following de novo assembly programs: Trinity (v2.5.1), SOAPdenovo2 (v2.4.1), Velvet (v1.2.10), and TransAbyss (v1.5.4) resulting in 4 ea individual assemblies [39,40,41,42]. The transcriptome assemblies were then aggregated together using EviGene, which deals with the many redundancies, picks the best representatives, and filters out misassemblies [43,44]. The reads were mapped to the assembly using NGen and quantified as RPKM using ArrayStar [45]. RPKMs for contigs identified by BLASTn of the assembly for key genes were extracted for each sample over all contigs matching that BLAST query. In addition, the transcriptome assembly was blasted with a query database created from NCBI scorpion toxin and our current sodium toxin databases (2133 total toxin sequences). From these Blast searches, RPKM values for the two males and female was summarized for sodium toxin RNAs with additional searches for additional scorpion toxin RNAs. The transcriptome datasets were deposited in NCBI with the following IDs: TSA: GIPT01000000, SRA: SRR11917465, BioProject: PRJNA636371, BioSample: SAMN15075759.

### 5.4. Toxin Proteomics

Telsons were harvested from flash frozen individual scorpions collected and maintained for the RT-qPCR analysis (four females and three males collected in NW AR), washed with 95% ethanol, and ground in lysis buffer (2% SDS in 100 mM Tris/HCl, pH 7.6). Extracted proteins were verified via gel electrophoresis on a 15% SDS-PAGE gel and sent to the UAMS Proteomics core laboratory for MS/MS spectrophotometry. Purified proteins were reduced, alkylated, and digested using filter-aided sample preparation with sequencing grade modified porcine trypsin (Promega) [46]. Tryptic peptides were then separated by reverse-phase XSelect CSH C18 2.5 um resin (Waters) on an in-line 150 × 0.075 mm^2^ column using an UltiMate 3000 RSLCnano system (Thermo). Peptides were eluted using a 90 min gradient from 97:3 to 60:40 buffer A:B ratio (Buffer A = 0.1% formic acid, 0.5% acetonitrile; Buffer B = 0.1% formic acid, 99.9% acetonitrile). Eluted peptides were ionized by electrospray (2.15 kV) followed by mass spectrometric analysis on an Orbitrap Fusion Lumos mass spectrometer (Thermo). MS data were acquired using the FTMS analyzer in profile mode at a resolution of 240,000 over a range of 375 to 1500 m/z. Following HCD activation, MS/MS data were acquired using the ion trap analyzer in centroid mode and normal mass range with precursor mass-dependent normalized collision energy between 28.0 and 31.0. Proteins were identified by database search using Mascot (Matrix Science) with a parent ion tolerance of 3 ppm and a fragment ion tolerance of 0.5 Da. Scaffold (Proteome Software) was used to verify MS/MS based peptide and protein identifications. Protein identifications were accepted if they could be established with less than 1.0% false discovery and contained at least 2 identified peptides. Protein probabilities were assigned by the Protein Prophet algorithm [47]. We further analyzed the proteomic data with blast searches against a fasta database of 3139 scorpion toxin sequences created from an NCBI search and our own sodium toxin diversity analyses. The mass spectrometry proteomics data have been deposited to the ProteomeXchange Consortium via the PRIDE [1] partner repository with the following dataset identifiers: PXD019851 (10.6019/PXD019851) and PXD019850 (10.6019/PXD019850).

## Figures and Tables

**Figure 1 toxins-13-00828-f001:**
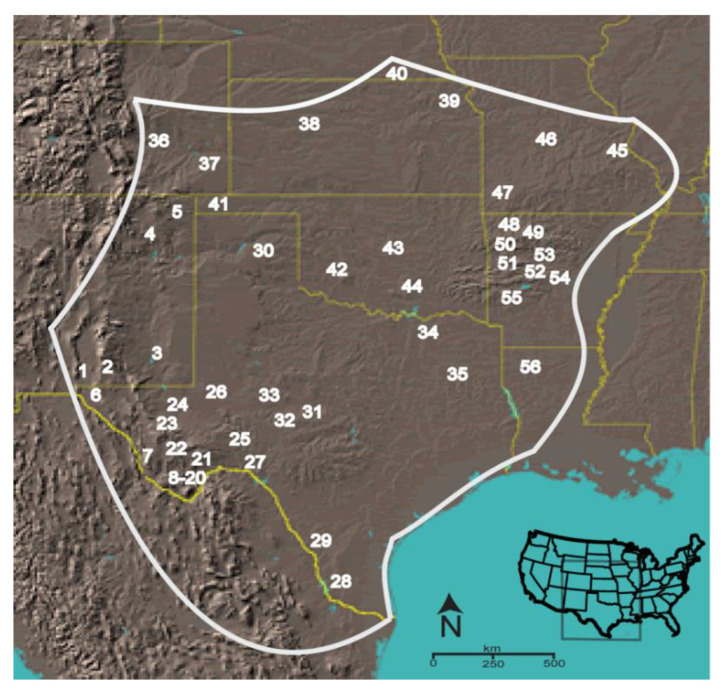
Approximate geographic range of *C. vittatus* with populations numbered as surveyed in a phylogeographic analysis [24]. The populations selected for the putative sodium β toxin gene diversity analysis are the following: (1) Aguirre Springs (AgSP), (6) Hueco Tanks (HT), (7) Chinati Hot Spring (Chin), (20) Boquillas Canyon Road (BCR), (30) PaloDuro (PaDu), (37) Lake Pueblo (LPb), (39) Lawrence (KS), (40) Little Blue River (LBR), (41) Black Mesa (BMe), (46) Ha Ha Tonka (Ha), (53) Scottsville (Scv), and (56) Kisatchie (Kisa).

**Figure 2 toxins-13-00828-f002:**
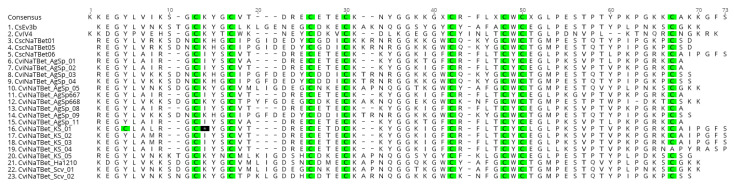
Select putative *C. vittatus* sodium β Toxin polypeptide sequences from western (AgSp) and eastern populations (KS, MO, AR) cataloged from a cloning survey of 10 population groups identified in a *C. vittatus* phylogeographic analysis. Also included are CsEv3b and CvIV4 toxin sequences and additional *C. sculupturatus* Beta toxin sequences obtained through additional cloning experiments. Cysteine residues are shown in green. The alignment for these sequences was conducted in Muscle (3.8.425) in the Geneious 11.1.5 package.

**Figure 3 toxins-13-00828-f003:**
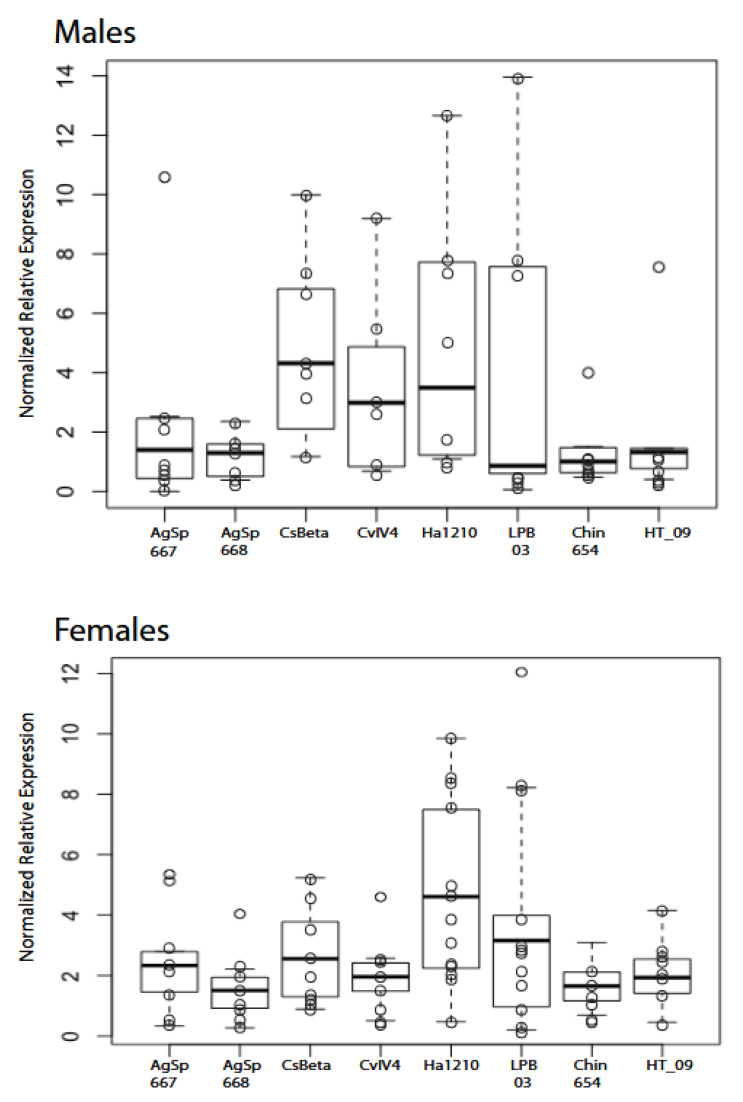
Box plots of normalized relative expression values for eastern population males (*n* = 8) and females (*n* = 13) with eight putative sodium β toxin genes. Normalized relative expression values (open circles for each individual) represent sodium β toxin genes adjusted to reference gene expression. The error bars represent SEMs. The only significant differences exist in females with Ha1210 to Chin654, AgSp668, and CvIV4 (see below).

**Table 1 toxins-13-00828-t001:** The five most common putative sodium β Toxins and their AA sequences identified across populations (98-100% similarity) from a cloning survey and ranked in order of common occurrence across populations. These AA sequences were identified in multiple populations.

CviNaTBet_HT_02 REGYLAIRGCIYSCVTDRECETECKKYGGKIGFCRFLTCYCEGLPKSVPTVPKPGRKCA
**8 populations:** HT, AgSp, Chin, LBR, LPb, PaDu, Kisa, KS
CviNaTBet_HT_01 KEGYLVNKSDGCKYGCVMLIGDEGCNKECKAPNQGGTKGWCYAFGCWCTGMPESTQVYPLPNKSCGKK
**6 populations:** HT, AgSp, LBR, LPb, PaDu, Scv
CviNaTBet_Chin_05 KEGYLVRKSDNCKHGCIPGIDEDFCDDICKARNRGGKKGWCQKYGCWCTGMPESTQTYPIPGKPCSS
**4 populations:** AgSp, Chin, BCR, Kisat
CviNaTBet_Chin_01 KEGYLVKKSDGCKYGCVMLIGDSNCDMECKAPNQGGQKGWCYAFGCWCTGMPESTQVYPLPGKSCGKK
**3 populations:** Chin, LBR, Kisat
CviNaTBet_HT_08 REGYLVNKKTGCKYGCTPKLGDDHCDKECKAPNQGGKKGWCKNFGCWCTGMPESTQTWPIPGKSCSS
**3 populations:** HT, AgSp, BCR

**Table 2 toxins-13-00828-t002:** Significant values between select putative sodium β toxins for Tukey’s pairwise test in females from normalized relative expression results in eight sodium β toxins and four reference genes. These three pairs represent significant values from 28 pairwise comparisons. Males did not exhibit significant differences among sodium beta toxin gene activity as determined by normalized relative expression values. See Appendix A for one-way ANOVAs and the complete Tukey’s pairwise test.

	Na Tox Gene	Diff lwr	upr	*p* adj
Ha1210-Chin654	2.958	0.619	5.297	0.0039
Ha1210-AgSp668	3.104	0.765	5.442	0.002
Ha1210-CvIV4	2.651	0.312	4.99	0.0149

**Table 3 toxins-13-00828-t003:** Transcriptome contig statistics from an assembly from three *C. vittatus* Telsons (Venom gland).

Assembled Contig Number	226,162
Assembled contigs > 500 bp	48,463
Assembled contigs > 1000 bp	26,909
Average length of assembled contigs (bp)	463
Longest assembled contig (bp)	9430
Total length (bp)	104,750,193
Assembled contig N50	1149

**Table 4 toxins-13-00828-t004:** RPKM values for Na toxins from a transcriptome contig assembly of *C. vittatus* body tissue (e.g., MC1-Male Carapace 1) and telson (venom gland, e.g., MT1-Male Telson 1) for two males and one female from a NW Arkansas population. Values for housekeeping and RT-qPCR reference genes are also shown. The six putative sodium β toxin genes in the first rows represent those targeted in the RT-qPCR experiments. The second putative sodium β toxin genes are additional unique transcripts identified after a BLAST query of the transcript to a scorpion toxin gene database. The last genes presented are commonly expressed genes and reference genes utilized in the RT-qPCR experiments. Accession numbers are provided for genes identified in the NCBI database.

	MC1	MT1	MC2	MT2	FC1	FT2
CviNaTBet_Ha1210	4	425	15	7638	4	4134
CviNaTBet_Chin654	5	502	17	8923	5	4932
CviNaTBet_AgSp667	3	523	12	6482	2	329
CviNaTBet_AgSp668	4	437	16	8682	4	3476
JF938594.1 Cv Alpha Tox IV4	0	224	1	406	0	199
AF338462.1 CsEv3b	4	365	21	11,318	4	1776
L05060.1_CUDNATOXA C_noxius						
Na channel blocker	4	363	21	11,278	4	1703
HQ262494.1 Css beta neurotox	1	356	4	2030	1	738
AF491130.1 C_limp Na channel Mod tox 5c	2	303	7	3976	1	165
AF491134.1 C_limp Na channel 8	2	227	8	4031	2	1411
AY164271.1 C_nox ergtox1	1	429	4	1830	1	469
AY159340.1 C_gracil ergtox1	0	59	0	43	0	54
AY649871.1 C_ex 13 tox	5	542	17	8941	5	4344
AY649868.1 C_ex10	2	151	8	4086	3	2657
AF338460.1 CsEIa	0	82	3	1471	1	1
AF338450.1 Cse8	4	386	22	12,095	4	1845
AF338448.1 CsE1x	2	268	9	4712	2	1565
AY649860.1 Cex2	4	360	21	11,258	4	1714
CviNaTBet_HT_04	2	121	11	6028	2	92
CviNaTBet_HT_08	3	176	10	5376	2	1597
CViNaTBet_HT_09	4	348	13	7181	4	3127
CviNaTBet_Kisa_01	2	200	9	4681	3	2544
CviNaTBet_LBR_03	2	161	8	3815	2	1944
CviNaTBet_LPb_04	2	122	8	3700	2	2255
CviNaTBet_Scv_02	3	139	12	6419	2	176
AF439766.1 C_limpidus 16s ribosomal	3392	2005	1894	1933	2204	1268
AH010433.1 CEX RNA poly	0	36	37	22	32	43
AY995829.1 C_nox COI	5951	2594	3837	1696	5481	2014
AY995849.1 C_Sc COII	2849	1251	1719	419	3458	1342
EU381110.1 C_vit COI	2357	1268	2393	547	2788	1352
CB334087.1 M_gib EF1	244	297	167	78	230	302
BU092001.1 M_gib EF2	1142	1103	721	369	881	944
CB334044.1 M_gib RPL19	255	260	147	105	273	244

**Table 5 toxins-13-00828-t005:** Sodium α and β toxin protein abundance as identified via Normalized Spectral Abundance Factor (NSAF) in seven adult scorpions telsons from a NW Arkansas population. The first three toxins are among those targeted in the RT-qPCR experiments. The complete list of identified toxin proteins with accession numbers is in Appendix A.

	Molecular Weight	Female 1	Female 2	Female 3	Female 4	Male 1	Male 2	Male 3
CviNaTBet_Ha1210	7 kDa	4.66%	15.08%	12.69%	10.03%	5.79%	5.66%	6.91%
CviNaTBet_AgSp668	7 kDa	1.09%	1.88%	0.58%	1.23%	na	na	0.35%
alpha-toxin, partial [*C. vittatus*]	11 kDa	0.75%	0.65%	1.21%	0.64%	0.81%	0.44%	0.73%
CviNaTBet_HT_01	7 kDa	2.15%	8.12%	5.48%	na	2.70%	na	na
CviNaTBet_HT_06	7 kDa	2.07%	2.94%	1.66%	1.05%	7.13%	3.62%	0.79%
CviNaTBet_Chin05	7 kDa	1.09%	1.88%	1.46%	1.23%	0.78%	3.51%	0.70%
CviNaTBet_BCR_10	7 kDa	1.09%	1.88%	1.46%	1.23%	0.78%	3.51%	0.70%
alpha-toxin CvIV4 [*C. sculpturatus*]	9 kDa	0.29%	0.95%	0.47%	0.25%	0.32%	0.52%	0.28%
RecName: Full = Toxin Cg2	8 kDa	na	na	0.87%	0.30%	na	na	0.35%
RecName: Full = Alpha-toxin Cn12	7 kDa	na	na	0.29%	0.31%	na	na	0.70%
beta-toxin Im-2-like [*C. sculpturatus*]	9 kDa	na	na	0.23%	na	na	na	0.55%
beta-toxin CeII8-like [*C. sculpturatus*]	10 kDa	na	na	0.23%	0.25%	na	na	na
RecName: Full = Alpha-toxin CsE5	7 kDa	na	na	0.31%	0.33%	na	na	0.37%
beta-neurotoxin CssIX precursor [*C. suffusus suffusus*]	9 kDa	na	na	0.24%	na	na	na	na

**Table 6 toxins-13-00828-t006:** A comparison among Buthid scorpion sodium channel toxin transcript and protein diversity as reported in this study and the literature. Alpha and Beta sodium variants were not separated in this Table.

Scorpion Species	Sodium Channel Toxin Transcript Diversity	Sodium Channel Toxin Protein Diversity	Reference
*Centruroides vittatus*	19	7	This paper
*Centruroides vittatus*	13	-	McElroy et al. 2017 [9]
*Centruroides hentzi*	36	9	Ward et al. 2018 [10]
*Centruroides sculpturatus **	22	-	Corona et al. 2001 [27]
*Centruroides sculpturatus **	-	5/6	Carcamo-Noriega et al. 2018 [6]
*Centruroides limpidus **	59	26	Cid-Uribe et al. 2019 [13]
*Centruroides noxius **	27	-	Rendón-Anaya et al. 2012 [17]
*Centruroides tecomanus **	24	30	Valdez-Velázquez et al. 2013 [28]
*Centruroides hirsutipalpus **	77	31	Valdez-Velázquez et al. 2020 [29]
*Tityus obscurus **	48	3	Oliveira et al. 2018 [30]
*Tityus serrulatus **	24	7	Oliveira et al. 2018 [30]
*Hottentotta gentili **	52	-	Grashof et al. 2019 [31]
*Androctonus mauretanicus **	42	-	Grashof et al. 2019 [31]
*Babycurus gigas*	13	-	Grashof et al. 2019 [31]
*Grosphus grandideri*	22	-	Grashof et al. 2019 [31]

* Medically important species are noted with an asterisk (*).

## Data Availability

All toxin DNA sequences were deposited in GenBank. Accession numbers are presented in Appendix A and Appendix A. The transcriptome datasets were deposited in NCBI with the following IDs: TSA: GIPT01000000, SRA: SRR11917465, BioProject: PRJNA636371, BioSample: SAMN15075759. The mass spectrometry proteomics data have been deposited to the ProteomeXchange Consortium via the PRIDE [1] partner repository with the following dataset identifiers: PXD019851 (10.6019/PXD019851) and PXD019850 (10.6019/PXD019850).

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
