# Peer review of "Reduced Toxicity of Centruroides vittatus (Say, 1821) May Result from Lowered Sodium β Toxin Gene Expression and Toxin Protein Production"

_toxins, 2021, doi:10.3390/toxins13110828_

Round 1

Reviewer 1 Report

The manuscript show interesting data about transcriptomic and proteomic analysis from  Centriroides vittatus, a north american scorpion. Besides that, the authors provide data about venom composition from other scorpion species indicating differences in the individual composition of the venom, mainly in . The authors showed that those differences are higher in sodium toxins from scorpion with higher medical significance than in scorpion with lower effects on humans.

Author Response

We thanks the reviewer for the comments on the manuscript.

Reviewer 2 Report

In this manuscript, the authors systematically analyzed the diversity of scorpion sodium beta-toxins on DNA, mRNA and protein levels in a population. Generally speaking, this is a very interesting story. The experiments were designed and performed well, also the manuscript was organized and written well. To move the manuscript forward to publishable level, couple of issues need to be addressed:

  1. The authors’ information is missing.
  2. To generate a phylogenetic tree based on the alignment shown in Figure 2 is going to help to deliver the information to the readers.
  3. The y-axis of the box charts in Figure 3 is wrong.
  4. The statement that “the toxin gene diversity is associated with the venom toxicity” doesn’t stand. I have no idea about the contribution of individual toxin to the overall toxicity. Basically, the toxicity of each toxin was not evaluated in this manuscript. In this case, let’s say scorpion A produces two toxins and one is super toxic; scorpion B produce ten toxins but none of them has impressive toxicity. In terms of the toxin diversity, B is better than A; but in terms of the overall toxicity, A may be better.

Reviewer 3 Report

For some reason your system does  not allows me to copy and paste may evaluation. I will send it in a separate file. Please see attached.

Author Response

Toxins-1337038  - Academic Editor Comments

There are several places that contain misconceptions about toxin evolution,

figure one is unclear,

We have clarified Figure 1

figure 2 is not an alignment (or if it is it is riddled with mistakes),

Figure two has been revised.

it is unclear how abundance was determined for table 1,

We have clarified table 1

the datapoints should be displayed in figure 3 (also the caption should say normalised to what),

We have revised figure three with additional datapoints and its figure title.

table 2 is a collection of tables that should be in the SI but results discussed in the main manuscript while b should be shown in its entirety and better explained in the caption,

We have revised Table two but kept the majority of the data in the supplementary table 2 as the data table is large.

table 3 show contig statistics not transcript statistics,

We have revised Table three with “assembled contigs” to clarify assembly rather than transcript.

the formatting on page 7 is off as are most of the other pages with tables,

We have reformatted the tables, but further assistance may be needed.

the rest of the results presented in table 5 should be shown so that the relative abundances of the other components are clear (important for interpreting the relative abundances shown),

We have revised Table five

table 6 needs to contain full genus names, the conclusion on line 333 is unwarranted unless the authors provide a clear measure on diversity (which is lacking in e.g., table 6). 

We have revised Table six and removed the conclusion on line 333.

In addition, the authors need to address the issues due to effects from the physiological states of the scorpions. I hope that I have missed it but as far as I understand, this was not controlled for and likely has a huge effect on toxin expression (recent use of venom or not, and for what) and relative protein abundance (how long since venom use and how well repleted were the glands). 

We have added information to outline the housing conditions and that all the scorpions represented resting venom extracted by telson removal and tissue disruption.

We have reviewed and edited the manuscript to clarify the results and the discussion.

We thank the reviewer’s comments to help improve this manuscript.
